# Biophysical Parameters of Plasma-Derived Extracellular Vesicles as Potential Biomarkers of Bone Disturbances in Breast Cancer Patients Receiving an Individualized Nutrition Intervention

**DOI:** 10.3390/nu15081963

**Published:** 2023-04-19

**Authors:** Carlos D. Coronado-Alvarado, Ana Teresa Limon-Miro, Herminia Mendivil-Alvarado, Jaime Lizardi-Mendoza, Elizabeth Carvajal-Millan, Rosa Olivia Méndez-Estrada, Humberto González-Ríos, Humberto Astiazaran-Garcia

**Affiliations:** 1Departamento de Nutrición y Metabolismo, Coordinación de Nutrición, CIAD, A.C., Hermosillo 83304, Mexico; ccoronado222@estudiantes.ciad.mx (C.D.C.-A.); limonmir@ualberta.ca (A.T.L.-M.); romendez@ciad.mx (R.O.M.-E.); 2Department of Medicine, University of Alberta, Edmonton, AB T6G 2R7, Canada; 3Coordinación de Tecnología de Alimentos de Origen Animal, CIAD, A.C., Hermosillo 83304, Mexico; jalim@ciad.mx (J.L.-M.); ecarvajal@ciad.mx (E.C.-M.); hugory@ciad.mx (H.G.-R.); 4Dpto de Ciencias Químico-Biológicas, Universidad de Sonora, Hermosillo 83000, Mexico

**Keywords:** exosomes, dynamic light scattering, breast cancer, nutrition intervention, bone mineral density

## Abstract

Extracellular vesicles (EVs) are implicated in several biological conditions, including bone metabolism disturbances in breast cancer patients (BCPs). These disorders hinder the adjustment of nutrition interventions due to changes in bone mineral density (BMD). The biophysical properties of EVs (e.g., size or electrostatic repulsion) affect their cellular uptake, however, their clinical relevance is unclear. In this study, we aimed to investigate the association between the biophysical properties of the plasma-derived EVs and BMDs in BCPs who received an individualized nutrition intervention during the first six months of antineoplastic treatment. As part of the nutritional assessment before and after the intervention, body composition including bone densitometry and plasma samples were obtained. In 16 BCPs, EVs were isolated using ExoQuick^®^ and their biophysical properties were analyzed using light-scattering techniques. We found that the average hydrodynamic diameter of large EVs was associated with femoral neck bone mineral content, lumbar spine BMD, and neoplasms’ molecular subtypes. These results provide evidence that EVs play a role in BCPs’ bone disorders and suggest that the biophysical properties of EVs may serve as potential nutritional biomarkers. Further studies are needed to evaluate EVs’ biophysical properties as potential nutritional biomarkers in a clinical context.

## 1. Introduction

Breast cancer has the highest incidence among all the types of cancers worldwide [1]. Adverse changes in body composition are frequently observed in breast cancer patients (BCPs) during antineoplastic treatment. These changes often include an increase in fat mass (FM) and a decrease in skeletal muscle mass and bone mineral density (BMD) [2,3,4], which can translate into adverse treatment effects and a worse prognosis [5,6,7]. Therefore, it is crucial to implement individualized nutrition interventions for all BCPs upon diagnosis to prevent unfavorable body composition changes related to the disease and treatment [8].

There is limited evidence regarding nutrition interventions that aim to protect bone health. In addition, biomarkers to assess early bone deterioration are scarce, especially in a clinical context. Therefore, bone disturbances in BCPs are usually detected in advanced stages of the disease that involve bone metastasis or established osteoporosis [9,10]. The deterioration of BCPs’ BMDs can be attributed to several factors, including cancer treatment-induced bone loss (CTIBL), body composition changes, and the molecular communication between breast cancer and bone cells [11,12]. Previously, FM was considered beneficial for BMD due to endocrine and mechanical factors; however, in recent years evidence has shown that body fat distribution, particularly increased abdominal and visceral fat, is negatively associated with BMD and inflammation has been linked to bone metabolism alterations [13]. On the other hand, the communication that breast cancer cells maintain with bone cells involves proteins, cytokines, and extracellular vesicles (EVs), but more specific emerging evidence is required [9,10]. This novel area of study in breast cancer should be further elucidated to better understand the disease progression and BMD deterioration in this population.

Extracellular vesicles are membranous particles that facilitate molecular communication between distant tissues through body fluids such as blood, saliva, and urine. These vesicles contain various types of molecular cargo, including non-coding RNA, proteins, and lipids [14]. Depending on their subcellular origin, EVs are classified into three main categories, each of which has a different range of size. These categories are apoptotic bodies (~50–5000 nm), microvesicles (~50–1000 nm), and exosomes (~30–150 nm) [15]. When the origins of EVs cannot be confirmed due to their isolation from a complex mix, it is recommended to classify them based on their size as either small EVs (<200 nm) or large EVs (>200 nm) [16]. Although their cellular uptake mechanism is not yet fully understood, EVs can be directed toward specific targets through surface modification [15]. It has also been suggested that the biophysical parameters of EVs, such as size, shape, electrostatic repulsion, and optical properties, are directly related to their surface characteristics and may impact their uptake process [17,18,19].

Recently, EVs have been proposed as potential biomarkers of many health and disease conditions and have been referred to as a promising tool for nutritional assessment [20]. In breast cancer research, EVs have been studied as drug delivery systems and as diagnostic tools. Several of their cargoes have been proposed as potential biomarkers for different conditions associated with this disease [15], including bone disturbances [9]. However, little is known about EVs’ biophysical parameters as potential biomarkers in this clinical context.

In this study, we aimed to evaluate the association between the biophysical properties of plasma-derived EVs measured by light-scattering techniques and BCPs’ BMDs at a baseline and after receiving an individualized six-month nutrition intervention during antineoplastic treatment [21,22]. We assessed the ζ potential as a measure of the electrostatic repulsion between EVs, the hydrodynamic diameter (*Dh*) as an estimate of the EVs size, and the relative intensity distribution (*%I*) of light reflected by large and small EV populations as an estimate of the relative abundance of these distinctively specific populations [18]. Studying the EVs’ biophysical properties is a step closer to better understanding how they can be related to other physiological variables in a clinical context and if they could aid in the adjustment of BCPs’ nutrition and medical interventions to prevent BMD deterioration.

## 2. Materials and Methods

### 2.1. Study Design and Subjects

A subsample was selected from the data and plasma samples collected in a quasi-experimental study that involved providing an individualized 6-month nutrition intervention to recently diagnosed BCPs under treatment [8,22]. The intervention was based on the dynamic macronutrient meal equivalent menu method customized for each individual according to recent evidence and dietary guidelines for BCPs previously described [8,21]. The original study was conducted between 2015 and 2018, during which BCPs underwent an integral nutrition assessment, including body composition, clinical, dietary, and blood sample collection, before starting medical treatment and after 6 months of the intervention. The selection criteria in the original study included recently diagnosed women with nonmetastatic breast cancer, free of known comorbidities, and previous antineoplastic treatment. Volunteers that developed metastasis or comorbidities during the study, used unsupervised nutrient supplementation, or voluntarily withdrew or stopped attending their respective treatment sessions were eliminated from the study. The research project was conducted in accordance with the Declaration of Helsinki. The protocol was approved by the Ethics Committee of the Research Center for Food and Development (CE/005/2015) and registered in ClinicalTrials.gov (NCT03625635). Further details about the original study, its results, and the intervention provided have been published elsewhere [8,21,22,23].

In this subsample study, the inclusion criteria required participants to have completed their participation and assessments in the original study, granted permission for storing their blood samples for further research, consented to these additional analyses, and had at least 1 mL of stored frozen plasma collected at the beginning and another one at the end of the intervention. The subsampling and analysis of data, including the isolation and assessment of EVs, were performed in 2021.

### 2.2. Blood Samples Management

In the original study, participants’ blood samples were collected after a 12-h fast prior to the start of the nutrition intervention and antineoplastic treatment, and again six months later. Blood samples were collected in a BD Vacutainer K2EDTA Tube using a BD vacutainer^®^ device. Immediately following collection, blood samples were centrifuged at 2500*× g* for 15 min at room temperature (Thermo Scientific^TM^ SL16R Centrifuge: TX-200 Swinging Bucket Rotor), to separate plasma. Samples were stored at −80 °C until the day we used them to isolate EVs.

### 2.3. Extracellular Vesicles Isolation

The EVs isolation process started by thawing each frozen plasma sample at 4 °C. Then, each sample was subjected to two rounds of centrifugation (SL16R Centrifuge: Microliter 30 × 2 mL Fixed Angle Rotor, Thermo Scientific^TM^, Waltham, MA, USA). The first centrifugation was performed at 10,000*× g* for 30 min at 4 °C; the supernatant was recovered and subjected to another centrifugation at 3000*× g* for 15 min at 4 °C. Next, 250 µL of the supernatant was recovered and 63 µL of ExoQuick^®^ (System Biosciences, Palo Alto, CA, USA) was added and each mixture was stored at 4 °C overnight. The following day, an EV pellet was obtained from every sample by centrifuging the mixture at 1500*× g* for 30 min at 4 °C (SL16R Centrifuge: Microliter 30 *×* 2 mL Fixed Angle Rotor, Thermo Scientific^TM^), discarding the supernatant and submitting the precipitate to an additional spin at 1500*× g* at 4 °C for 15 min, and discarding the supernatant once again. The pellet was resuspended in 250 µL of phosphate-buffered saline solution and then it was sonicated (250 Sonifier, Branson Ultrasonics^TM^, Brookfield, CT, USA) at a 30% duty cycle for 20 s. At this point, each sample was immediately analyzed by light-scattering techniques [18].

Additionally, pooled plasma was prepared from each volunteer’s frozen plasma sample after thawing them at 4 °C. The EVs were isolated from the pooled plasma by following the same protocol as mentioned before, except that this time ExoQuick^®^ ULTRA (System Biosciences) was used instead of ExoQuick^®^ with the additional steps recommended by the manufacturer. The EVs isolated from the pooled sample were assessed by an antibody array and Transmission Electron Microscopy (TEM) to complete the EVs’ characterization [16].

### 2.4. Light-Scattering Techniques

The ζ potential and the average *Dh* of the particles in the suspension of each plasma-derived EV sample were analyzed using electrophoretic and dynamic light scattering, respectively. The measurements were conducted using Möbius™ (Wyatt Technology, Santa Barbara, CA, USA) equipment set at 37 °C to mimic body temperature. The software DYNAMICS^®^ (Wyatt Technology) was used to analyze the intensity-weighted distribution of the suspended particles by applying filtering criteria to define two populations of particles: small EVs with a hydrodynamic radius between 10 and 100 nm and large EVs with a radius from 100 to 500 nm. The *Dh* of each population and their corresponding *%I* were then analyzed.

### 2.5. Antibody Array

An antibody array was performed to further characterize the pooled, plasma-derived EVs. For this purpose, ExoCheck^®^ (System Biosciences) was used following the manufacturer’s instructions. Luminol-peroxidase was used as the developing agent for the array.

### 2.6. Transmission Electron Microscopy (TEM)

The pooled, plasma-derived EVs sample was fixed with 2.5% glutaraldehyde in sodium cacodylate solution and adsorbed onto a formvar-coated (0.3%) copper grid. The grid was then stained with 2.5% uranyl acetate for 30 s and then dried at room temperature. Once the grids were dried, they were assessed through a JEM-1011 transmission electron microscope (JEOL Ltd., Tokyo, Japan).

### 2.7. Bone Densitometry and Body Composition Assessment

During the original study, bone densitometry and body composition assessments were carried out by dual-energy X-ray absorptiometry (DXA) using a Discovery WI (QDR SERIES) densitometer (Hologic, Waltham, USA) at the beginning and again at the end of the intervention [22]. Total body scans were used to assess total FM, abdominal fat, total appendicular skeletal muscle mass (TASM), BMD, and bone mineral content (BMC). To standardize the FM, it was divided by the square of the participant’s height to obtain their Fat Mass Index (FMI), and the same was applied to the TASM to obtain their Appendicular Skeletal Muscle Mass Index (ASMI) [24]. Scans of the spine, specifically from the L1 to L4 vertebral bodies, were used to assess lumbar BMC, lumbar BMD, lumbar T-score, and lumbar Z-Score. Proximal femur scans were performed to obtain total hip BMC, total hip BMD, total hip T-score, total hip Z-score, femoral neck BMD, femoral neck BMC, femoral neck T-score, and femoral neck Z-score.

### 2.8. Statistical Analysis

Descriptive analyses were performed to report frequencies and proportions as well as continuous data as means ± standard deviations (SD) and nonparametric data as median and interquartile ranges (IQR). The Wilcoxon signed-rank test was used to compare the changes in body composition and the biophysical parameters of plasma-derived EVs after the first six months of intervention and antineoplastic treatment. The same was applied for changes in bone densitometry after stratifying by menopausal state.

To determine the associations between the biophysical parameters of plasma-derived EVs, bone densitometry, and clinical features while adjusting for possible confounders, mixed-effects linear regression models were developed as follows: bivariate models between variables of interest and possible predictors were calculated and filtered through both plausibility and statistical (*p* ≤ 0.2) criteria. A stepwise approach was then applied (entrance *p* value ≤0.05; removal *p* value > 0.05) to acquire preliminary models that then were assessed for possible interactions and correlations between their predictors. Finally, assumptions of the models were evaluated with residual plot graphics. In some cases, the sensitivity of the models was assessed by fitting variables of biological importance into the final models even if rejected through the modeling method. All the statistical analyses were carried out in STATA 15 and statistical significance was considered on a two-tailed *p*-value below 0.05.

## 3. Results

Of the 36 BCPs that were enrolled in the original study, 22 completed the nutritional intervention. Of those, 16 were admitted to the subsample analyses, and 6 were left out due to insufficient plasma samples stored for EVs analyses (Figure 1).

Subsample participants’ mean age was 39 ± 5 years old for the premenopausal group (*n* = 9) and 61 ± 5 years old for the postmenopausal group (*n* = 7). The most common molecular type of breast tumor was Luminal A and B (*n* = 7, respectively). During the follow-up, two of the BCPs did not receive any cycle of chemotherapy, and three did not receive any radiation therapy. All patients underwent surgical treatment, five having a quadrantectomy and ten having a mastectomy, one of which was bilateral. Further information regarding baseline clinical features can be found in Table 1.

### 3.1. Changes in the Subsample after the Nutrition Intervention

All the BCPs in the subsample showed improvement (*p* < 0.05) in behaviors related to bone health (i.e., hours of sleep and physical activity) and in adiposity at all levels after the nutrition intervention. On the other hand, appendicular skeletal muscle mass was preserved as there were no changes in the ASMI (*p* > 0.05). According to the body mass index (BMI), there was a healthy shift when comparing the baseline and six-month anthropometric measurements after the nutrition intervention. At the beginning of the study, six participants were classified as having normal body weight (37.6%), three were overweight (18.7%), and seven were in the obesity classification (43.7%). After the intervention, all the normal-weight BCPs in the subsample remained in that category and all the BCPs with obesity or overweight reduced their body weight, resulting in seven within a normal-weight (43.8%) range, five with overweight (31.2%), and four with obesity (25%).

Despite these beneficial changes, BCPs in this subsample presented detrimental alterations in their bone densitometry (Table 2). In premenopausal women, it was observed that the lumbar BMD and total hip BMD were lower (*p* < 0.05) after the first six months of antineoplastic treatment. On the other hand, postmenopausal women had less femoral neck BMD and total hip BMD (*p* < 0.05) after the six-month follow-up when compared to their baseline assessments. According to the T-scores of the three regions at the baseline of the original study, only three of the postmenopausal BCPs in the subsample had an acceptable BMD, three had osteopenia, and one had osteoporosis. None of them were aware of having a low BMD. After the six-month follow-up, all the postmenopausal BCPs had a reduction in their T-scores in at least one region, leading to four of them being in the osteopenia classification and two presenting osteoporosis.

### 3.2. Biophysical Properties of Plasma-Derived EVs

The median ζ potential of the plasma-derived EVs at the baseline was −8.6± mV, with an IQR of 2 mV, and after six months it was −8.6 mV with an IQR of 1.6 mV. The median *Dh* of the small plasma-derived EVs population and their *%I* was 90.7 nm (IQR of 50.5 nm) and 36.8% (IQR of 37.9%) at the baseline. After six months, these values were 100 nm (IQR 38.9 nm) and 31.1% (IQR 53.3%), respectively. In the case of the large plasma-derived EVs, the median *Dh* and median *%I* were 500.5 nm (IQR of 374.9 nm) and 29.4% (IQR of 57%) at the baseline and 500.1 nm (IQR of 273.9 nm) and 31.1% (IQR of 53.3%) after six months. However, none of those were statistically-significant changes (*p* > 0.05). The mean time of storage for the frozen plasma was 53.6 ± 6 months for samples drawn at the baseline and 47.3 ± 6 months for those drawn at the end of the intervention. An example of the regularization graphs obtained by DLS showing both subpopulations of the particles and TEM images can be seen in Appendix A.

### 3.3. Associations between Bone Densitometry and Biophysical Properties of Plasma-Derived EVs

Each of the 14 bone densitometry parameters studied was used as a dependent variable to create mixed-effect linear regression models. Four of them were statistically associated with at least one biophysical property of plasma-derived EVs (Table 3). The *Dh* of the large plasma-derived EVs was ambiguously associated with the spine BMD and the femoral neck BMC, behaving like a protective factor in the first association and as a risk factor in the second one. For its part, the *%I* of large plasma-derived EVs was negatively associated with the lumbar Z-score and total hip BMD. We did not find any association between the *Dh* or *%I* of small plasma-derived EVs population and bone densitometry parameters, nor did the ζ potential.

### 3.4. Associations between Biophysical Properties of Plasma-Derived EVs and Neoplasm Characteristics

In addition to evaluating the associations between the bone densitometry and biophysical properties of plasma-derived EVs, we also explored the relationship between these properties and the characteristics of the neoplasms in the BCPs’ subsamples (Table 4). We found that the *Dh* of both the plasma-derived EVs populations were associated with the molecular types of the breast tumors. The *Dh* of small EVs were smaller in HER2/Neu in comparison to the triple negative. The *Dh* of large EVs were also smaller in HER2/Neu, followed by the Luminal B and Luminal A subtypes in increasing order when comparing them with the triple-negative subtype, respectively. The ζ potential was associated with the overexpression of the proteins HER2 and Ki-67 in BCPs’ tumors.

## 4. Discussion

In this study, we evaluated the association of the physical properties of plasma-derived EVs with bone densitometry parameters in BCPs at diagnosis and after six months of antineoplastic treatment and an individualized nutrition intervention. To our knowledge, this is the first study to evaluate the relationship between the biophysical properties of plasma-derived EVs assessed with light scattering techniques and the BMD in BCPs receiving an individualized nutrition intervention. As expected, the findings in the subsample regarding behaviors related to health, body composition, and bone densitometry were similar to those found in the whole sample of the original study, including a significant bone loss in both pre- and postmenopausal women [22].

It has been widely reported that cancer treatments lead to bone loss, particularly those that interfere with estrogen metabolism [25]. The most affected anatomical site for the premenopausal women in this subsample was the lumbar spine, having a median change of −3.07% of lumbar BMD after six months of antineoplastic treatment, similar to what has been reported before [26]. This rate of bone loss is more than twice the expected annual bone loss in healthy premenopausal women at this site [27]. In postmenopausal women, the proximal femur was the most affected site, with a median change of −4.8% of femoral neck BMD after six months of antineoplastic treatment. The expected annual bone loss in healthy postmenopausal women at this site is 1.4% [27]. It is important to note that a limitation of this study was the lack of information about the precise regimen of drugs used by the BCPs in the subsample. However, most of them had been diagnosed with luminal tumors, with it being unlikely that they did not receive any antiestrogenic treatment.

According to the mixed-effects linear regressions, another factor that was associated with the changes in BMD in the participants of this study was the *Dh* and *%I* of their large plasma-derived EVs. While the precise impact of EV size on function remains unclear, there is a diverse biological classification of EVs with different functions that vary in their size ranges [28]. Within bone metabolism, a special type of EVs called matrix vesicles (MtVs) plays an important role. Small MtVs start the bone mineralization process, but the large ones maintain long-term secondary mineralization [29]. More studies are needed to evaluate the possibility of detecting large MtVs in BCPs’ plasma in relation to their bone disturbances.

Oncosomes are a type of EV whose size plays a crucial role as they contain cancer-related cargo and originate from the plasma membrane of cancer cells [28,30]. Our study revealed that the size of large plasma-derived EVs varied depending on the molecular subtype of the neoplasm. Similarly, this variation was observed for small plasma-derived EVs but was only significant between the HER2/Neu and triple-negative tumors, although the lack of significance in Luminal A and Luminal B tumors could be due to the small sample size. In both populations, plasma-derived EVs were larger in triple-negative tumors. However, Li et al. reported contradictory findings where small EVs derived from the plasma of triple-negative BCPs were smaller than those with HER2/Neu, proving a method to infer EVs sizes from their membrane viscosity [31]. These differences could be explained by the small size of the samples or the use of different techniques. The heterogeneity of EVs poses a significant challenge when studying specific populations [16,18,32]. This is particularly relevant in cancer contexts as the disease is linked to a broader diversity of EVs [30,33]. Therefore, standardized methods are required to assess EV populations and their biophysical properties for comparing results and understanding their function and clinical relevance better.

The relationship between the size of EVs and their cargo is not arbitrary. Kim et al. designed a system to purify EV populations based on their size, detecting a greater proportion of nucleic acids in large EVs’ cargo when compared to other populations [34]. This is interesting as miRNAs have been proposed as a possible cargo of circulating EVs that could be involved in the development of bone metastasis in BCPs [9]. Our research findings indicate that large plasma-derived EVs play a paradoxical role in bone loss in different anatomical sites; therefore, studies focused on their cargo are needed.

Another association we discovered in our volunteers was that the ζ potential changes if there is an overexpression of the HER2 and/or Ki-67 proteins. The ζ potential in biological membranes largely depends on its composition [35]. It has been stated that the unsaturated fatty acids profile within breast cancer cell membranes changes if the cell overexpresses HER2 or Ki-67 compared to those that do not [36,37]. The amount of unsaturated fatty acids, particularly omega-3, can alter the protein expression on cancer cell membranes which could affect their ζ potential [38]. Interestingly, this association was significant only when adjusted by abdominal fat mass. Body fat distribution can affect lipid metabolism, which reflects in the membrane composition [39]. Nonetheless, further studies are needed to confirm whether membrane protein changes due to the lipid profile variations significantly alter the ζ potential of cells and of the EVs they produce and if these changes could be related to abdominal fat deposits.

This study has several limitations, including its design and sample size; therefore, these results should be interpreted carefully. The main limitation was that specific details of the antineoplastic treatment were not available to assess as a possible confounder. Moreover, it has been reported that the storage method after the EVs’ collection could affect their shape and colloidal stability, which could change their ζ potential and size [40]. Nonetheless, more studies are needed to understand if these changes occur in EVs isolated from frozen plasma. Future studies with a larger sample size should be designed to evaluate and comprehend the role that the biophysical properties of plasma-derived EVs play on bone disturbances in BCPs.

## 5. Conclusions

The biophysical properties of extracellular vesicles can be related to other physiological variables, which implies that they could play an important role in their functionality. These properties should be assessed as part of extracellular vesicles’ clinical research. The large plasma-derived extracellular vesicles’ population size was associated with changes in bone mineral density and with the molecular type of the tumor in breast cancer patients. If these findings are validated in larger studies in the future, extracellular vesicles could become a useful tool to evaluate early changes in breast cancer patients’ bone health, which would improve the prevention and treatment of osteoporosis, osteopenia, and health outcomes from both clinical nutrition and medical perspectives.

## Figures and Tables

**Figure 1 nutrients-15-01963-f001:**
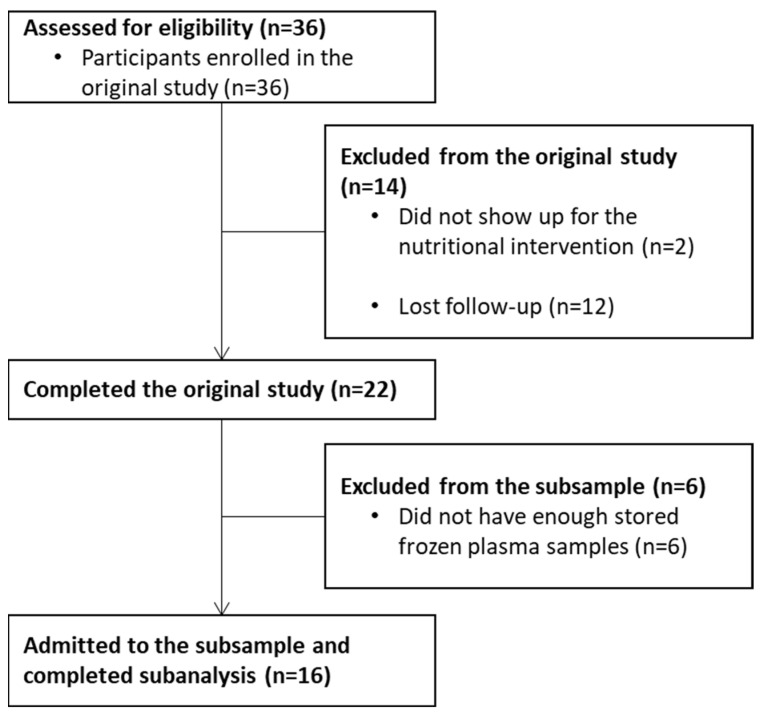
CONSORT flow diagram of the subsampling process.

**Table 1 nutrients-15-01963-t001:** Clinical characteristics of the breast cancer patients admitted to the subsample analyses.

Clinical Characteristics	Premenopausal BCPs (*n* = 9)	Postmenopausal BCPs (*n* = 7)
Age (years old)	39 ± 5	61 ± 5
Body weight (kg)	75.5 ± 16	68.7 ± 14
Body Mass Index (kg/m^2^)	29.3 ± 6	26.7 ± 4
Normal weight (*n*, (%))	2 (22.2)	4 (57.1)
Overweight (*n*, (%))	3 (33.3)	0 (0)
Obese (*n*, (%))	4 (44.4)	3 (42.9)
Fat Mass Index (kg/m^2^)	12.1 ± 3	10.7 ± 3
Abdominal fat (kg)	2.8 ± 1	2.4 ± 2
ASMI (kg/m^2^)	5.6 ± 1	4.9 ± 1
Physical activity level according to IPAQ (*n*, (%))		
Low physical activity level	5 (55.6)	5 (71.4)
Moderate physical activity level	4 (44.4)	2 (28.6)
Tumor grade (*n*, (%))		
Grade 1	1 (11.1)	1 (14.3)
Grade 2	6 (66.7)	6 (85.7)
Grade 3	2 (22.2)	0 (0)
Molecular type of tumor (*n*, (%))		
Luminal A	3 (33.3)	4 (57.1)
Luminal B	4 (44.4)	2 (28.6)
HER2/Neu	1 (11.1)	1 (14.3)
Triple negative	1 (11.1)	0 (0)

Quantitative variables are expressed as mean ± standard deviations. ASMI: Appendicular Skeletal Muscle Mass Index. BCPs: Breast Cancer Patients. IPAQ: International Physical Activity Questionnaire.

**Table 2 nutrients-15-01963-t002:** Changes in bone densitometry of breast cancer patients in the subsample before and after six months of antineoplastic treatment and nutrition intervention.

Anatomical Regions	Premenopausal BCPs (*n* = 9)			Postmenopausal BCPs (*n* = 7)		
Bone Densitometry Parameters	Before	After	*p* *	Before	After	*p* *
**Total body scan**						
BMC (g)	2078.7 (487.5)	2119.2 (457.9)	0.95	2001.2 (613.3)	1989.4 (810.6)	0.31
BMD (g/cm^2^)	1.17 (0.10)	1.17 (0.11)	0.67	1.18 (0.38)	1.17 (0.27)	1.00
**Lumbar spine scan**						
BMC (g)	54.80 (9.91)	57.92 (1.90)	0.95	48.70 (11.57)	51.13 (10.97)	1.00
BMD (g/cm^2^)	1.02 (0.11)	0.98 (0.09)	0.02	0.96 (0.29)	0.914 (0.25)	0.06
Z-score (SD)	0.10 (1.20)	-0.20 (1.00)	0.02	0.70 (1.60)	0.60 (1.90)	0.12
T-score (SD)	-	-	-	−0.70 (1.90)	−1.20 (2.40)	0.14
**Proximal femur scan**						
Total hip BMC (g)	34.42 (4.24)	33.74 (5.98)	0.76	26.63 (7.97)	26.95 (6.45)	0.86
Total hip BMD (g/cm^2^)	1.02 (0.08)	1.00 (0.05)	0.01	0.82 (0.15)	0.80 (0.12)	0.04
Total hip Z-score (SD)	0.50 (0.06)	0.60 (0.50)	0.14	0.10 (2.10)	0.00 (1.00)	0.04
Total hip T-score (SD)	-	-	-	−1.00 (1.20)	−1.10 (0.90)	0.04
Femoral neck BMC (g)	3.50 (0.94)	3.95 (0.54)	0.28	2.83 (1.06)	2.70 (0.86)	0.12
Femoral neck BMD (g/cm^2^)	0.88 (0.25)	0.16 (0.28)	0.28	0.69 (0.13)	0.69 (0.13)	0.02
Femoral neck Z-score (SD)	0.20 (1.50)	1.50 (0.33)	0.33	0.00 (1.10)	−0.30 (0.80)	0.04
Femoral neck T-score (SD)	-	-	-	−1.50 (1.2)	−1.50 (0.80)	0.34

Values are expressed as medians and interquartile ranges in parentheses. BCPs: Breast Cancer Patients. BMC: Bone Mineral Content. BMD: Bone Mineral Density. * Calculated with the Wilcoxon signed-rank test.

**Table 3 nutrients-15-01963-t003:** Mixed-effect linear regression models for prediction of bone densitometry parameters in breast cancer patients receiving a nutritional intervention during the first six months of antineoplastic treatment.

Predicted Bone Densitometry Parameter	Model 1 *		Model 2 **	
Explanatory Variables	β	*p*	β	*p*
**Lumbar spine BMD (g/cm^2^)**				
Large plasma-derived EVs *Dh* (nm)	0.00007	0.01	0.0001	<0.01
Body weight (kg)	0.00637	<0.01	0.006	<0.01
Age (years old)	-		−0.003	0.63
Menopausal status				
Premenopausal	-		Ref.	
Postmenopausal	-		0.035	0.78
**Lumbar spine Z-score (SD)**				
Large plasma-derived EVs *%I* (%)	−0.00449	<0.01	−0.00461	<0.01
Fat Mass Index (kg/m^2^)	0.15649	<0.01	0.16050	<0.01
Age (years old)	-		−0.03274	0.50
Menopausal status				
Premenopausal	-		Ref.	
Postmenopausal	-		1.404	0.22
**Femoral neck BMC (g)**				
Large plasma-derived EVs *Dh* (nm)	−0.00074	<0.01	−0.00072	<0.01
Age (years old)	−0.05672	<0.01	−0.08105	0.03
Coffee consumption (cups/day)	0.15564	<0.01	0.08087	0.09
Body weight (kg)	-		0.02871	0.01
Menopausal status				
Premenopausal	-		Ref.	
Postmenopausal	-		0.93931	0.30
**Total hip BMD (g/cm^2^)**				
Large plasma-derived EVs *%I* (%)	−0.00021	<0.01	−0.00019	0.02
Time of measurement				
Before treatment	Ref.		Ref.	
Six months of treatment	−0.01758	<0.01	−0.015513	<0.01
Menopausal status				
Premenopausal	Ref.		Ref.	
Postmenopausal	−0.15982	<0.01	−0.20204	0.05
Age (years old)	-		0.00237	0.26
Body weight (kg)	-		0.00117	<0.01

* Model generated by the stepwise method. ** Model generated by fitting additional variables of biological importance originally rejected during the stepwise method. BMC: Bone Mineral Content. BMD: Bone Mineral Density. *Dh*: Hydrodynamic Diameter. EVs: Extracellular Vesicles. %*I*: Relative Intensity Distribution.

**Table 4 nutrients-15-01963-t004:** Mixed-effect linear regression models for prediction of biophysical properties of plasma-derived extracellular vesicles in breast cancer patients receiving an individualized nutrition intervention during the first six months of antineoplastic treatment.

Predicted Biophysical Property	Model 1 *		Model 2 **	
Explanatory Variables	β	*p*	β	*p*
**Large plasma-derived EVs *Dh* (nm)**				
Molecular type of tumor				
Luminal A	−417.67	<0.01	−438.68	<0.01
Luminal B	−336.86	0.02	−331.04	0.04
HER2/Neu	−485.90	<0.01	−480.47	<0.01
Triple negative	Ref.		Ref.	
Physical activity (min/week)	0.44	0.03	0.43	0.06
Storage time (months)	-		0.00	0.99
Temperature at measurement (°C)	-		0.54	0.58
**Small plasma-derived EVs *Dh* (nm)**				
Molecular type of tumor				
HER2/Neu	−44.55	0.04	−46.17	0.06
Triple negative	Ref.		Ref.	
Storage time (months)	-		−0.65	0.60
Temperature at measurement (°C)	-		2.81	0.97
**ζ potential (mV)**				
HER2 status				
Overexpressed	−1.59	<0.01	−1.68	<0.01
Normally expressed	Ref.		Ref.	
Ki-67				
≤15%	Ref.		Ref.	
>15%	−0.93	0.02	−1.15	0.02
Abdominal fat (kg)	0.42	0.01	0.42	0.01
Storage time (months)	-		0.03	0.39
Temperature at measurement (°C)	-		−2.19	0.43

* Model generated by the stepwise method. ** Model generated by fitting additional variables of biological importance originally rejected during the stepwise method. *Dh*: Hydrodynamic Diameter. EVs: Extracellular Vesicles.

## Data Availability

The data presented in this study are available on request from the corresponding author. The data are not publicly available due to privacy reasons.

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
