# Peer review of "Biophysical Parameters of Plasma-Derived Extracellular Vesicles as Potential Biomarkers of Bone Disturbances in Breast Cancer Patients Receiving an Individualized Nutrition Intervention"

_nutrients, 2023, doi:10.3390/nu15081963_

Round 1

Reviewer 1 Report

In the present manuscript entitled “Biophysical parameters of plasma-derived extracellular vesicles as potential biomarkers of bone disturbances in breast cancer patients receiving an individualized nutrition intervention,” the authors made a good attempt to discover the potential biomarker for bone disturbances in breast cancer. My specific comments are below.

1.      The authors mentioned that the patients received antineoplastic treatment, so it is better to provide what kind of treatment was given to them (targeted therapy, immunotherapy. etc).

2.      If possible, authors could provide TEM and Particle size images in supplementary materials or the main manuscript.

3.      Authors are advised to elaborate on the introduction section regarding exosomes (like size, and parameters) here is the reference article, and authors can cite https://doi.org/10.3390/cancers14061435

Reviewer 2 Report

This paper evaluated the association between the biophysical properties of EVs and the bone mineral density of breast cancer patients accepted the individualized nutrition intervention. There are some questions to answer about this paper:

1. Although you have listed the limitations of this study, the cohort just contained 16 BCPs plasma samples are too small and the results obtained from your paper is unpersuasive. You should supplement some experiments or literatures to verify the reasonability of your research.

2. What’s the contents or objectives of the “original study”? If there are similar contents between this paper and the “original study”?

3. What’s the “nutrition intervention” for each breast cancer patient? If the changes of the biophysical parameters of EVs and the BMD are resulted from different treatment project rather than the progress of breast cancer?

4. You should elucidate the situation of Dh in small EVs more particularly.

5. Please listed the equation of the mixed-effect linear regression models and show the independent variable and dependent variable. How we can use theses models to predict the bone densitometry parameters after the antineoplastic treatment ? Please elucidate the process of prediction in detail.

6. In your study, the clinical features are unfixed, such as the age, cancer type, surgery type and the treatment ways. So how do we know if these factors were involved in the relationship between BMD and the biophysical parameters of EVs? The authors might be skeptical about your results.

7. Your language should be polished.

Reviewer 3 Report

In the present article authors have investigated the involvement of extracellular vesicles in bone disturbances of breast cancer patients. Bone density loss and body composition change with antineoplastic treatment is a major concern. Nutritional intervention catered to bone protection added to breast cancer patients’ treatment regimen at earlier stages is essential. 

Did authors consider of having an animal model to study this is more detail? May be some of those limitations can be met with animal models. Even, EVs isolation can be optimized through animal models. 

Reviewer 4 Report

This manuscript entitled "Biophysical parameters of plasma-derived extracellular vesicles as potential biomarkers of bone disturbances in breast cancer patients receiving an individualized nutrition intervention" aimed to explore the association between bone mineral density (BMD) and the biophysical properties of extracellular vesicles (EVs) in breast cancer patients (BCPs) in which patients received nutritional intervention during the first six months of antineoplastic treatment and nutritional condition, BMD , as well as EVs physical properties were analyzed before and after intervention then the data were compared. Results showed nutrition intervention would improve bone health and large EVs’ hydrodynamic diameter were associated with femoral neck bone mineral content, lumbar spine BMD, and molecular subtype of neoplasms. Authors concluded EVs are involved in BCPs bone disturbances and their biophysical properties would be potential as biomarkers of bone health.

Overall, exploring the association between BMD and the biophysical properties of plasma-derived EVs in the BCPs is interesting and valuable for academic and clinical reference. However, there are some should be clarified or deeply discussed.

1.     In the abstract (L27-28), “Clinical relevance of EVs biophysical properties should be further evaluated as possible nutritional biomarkers” are EVs potential nutritional biomarkers? This study just elucidated association between BMD and the biophysical properties of plasma-derived EVs but not nutritional condition; why are EVs biophysical properties further evaluated as possible nutritional biomarkers? What is the meaning and significance of nutritional biomarkers?

2.     How are EVs defined? What are the classes and components of EVs?

3.     May nutritional intervention affect the contents of EVs among different class?

Round 2

Reviewer 2 Report

This paper could be accepted.